# Unleashing the Potential of Multi-Channel Fusion in Retrieval for Personalized Recommendations

## ABSTRACT

Recommender systems (RS) are pivotal in managing information overload in modern digital services. A key challenge in RS is efficiently processing vast item pools to deliver highly personalized recommendations under strict latency constraints. Multi-stage cascade ranking addresses this by employing computationally efficient retrieval methods to cover diverse user interests, followed by more precise ranking models to refine the results. In the retrieval stage, multi-channel retrieval is often used to generate distinct item subsets from different candidate generators, leveraging the complementary strengths of these methods to maximize coverage. However, forwarding all retrieved items overwhelms downstream rankers, necessitating truncation. Despite advancements in individual retrieval methods, **multi-channel fusion**, the process of efficiently merging multi-channel retrieval results, remains underexplored. **We are the first to identify and systematically investigate multi-channel fusion in the retrieval stage.** Current industry practices often rely on heuristic approaches and manual designs, which often lead to suboptimal performance. Moreover, traditional gradient-based methods like SGD are unsuitable for this task due to the non-differentiable nature of the selection process. In this paper, we explore advanced channel fusion strategies by assigning systematically optimized weights to each channel. We utilize black-box optimization techniques, including the Cross Entropy Method and Bayesian Optimization for global weight optimization, alongside policy gradient-based approaches for personalized merging. Our methods enhance both personalization and flexibility, achieving significant performance improvements across multiple datasets and yielding substantial gains in real-world deployments, offering a scalable solution for optimizing multi-channel fusion in retrieval.

## 1 INTRODUCTION

In the era of information overload, recommender systems (RS) have become indispensable in modern web services, ranging from video streaming platforms to online shopping services. One of the main technical challenges in RS is to efficiently process billions of items to provide personalized experiences to millions of users under *strict latency restrictions* [25, 34]. As shown in Figure 1, a widely used solution is multi-stage cascade ranking systems [11, 14, 60]. In the first stage of the cascade system, known as the retrieval stage (also called matching or recall stage [43, 70]), a group of computationally efficient candidate generators selects a small set of candidates. These candidates are then further filtered, ranked, and ultimately presented to the user by slower but more accurate rankers. In this process, the retrieval stage acts as both the cornerstone and bottleneck of the RS. Without effective retrieval, even the most advanced ranking algorithms cannot perform optimally.

Typically, multi-channel retrieval [37] is essential for efficiently and effectively retrieving items from large-scale item pools, as shown in Figure 1. Top-$K$ items from each channel are merged

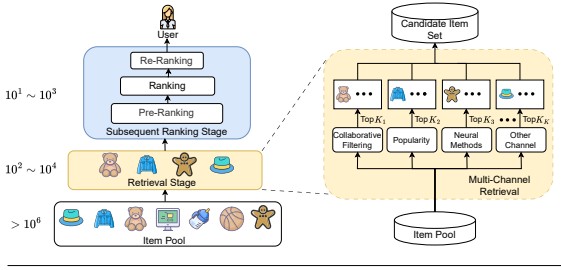

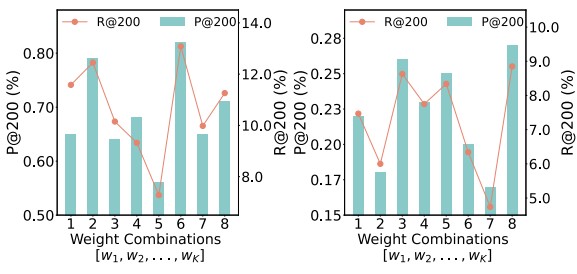

**Figure 1: Up: Illustration of multi-stage cascade ranking and multi-channel retrieval in recommender systems. Bottom: Performance variations with different weight combinations on Gowalla (left) and Amazon_Books (right).**

and passed to the next stage, with $K$ varying across channels. The underlying reason is that forwarding all retrieved items from each channel would overwhelm downstream rankers, necessitating truncation. Therefore, the primary challenge in multi-channel retrieval lies in effectively merging the diverse items retrieved by each candidate generator. This process involves determining the appropriate $K$ for each channel's top-$K$ selection or assigning optimal weights to each channel during the merging process. For more details on why multi-channel retrieval is favored over single-channel and the rationale behind weight assignment, refer to Sections 2 and 3.2.

Despite advancements in individual retrieval methods [11, 23, 36], the task of efficiently merging multi-channel retrieval results, which we define as **multi-channel fusion**, has received limited attention. We identify three key challenges in multi-channel fusion:

- **(C1)** Current industry practices often rely on heuristic approaches and manual designs, such as snake-merge or simple quota mechanisms [37], guided by business needs. These methods lack systematic analysis, leading to suboptimal performance and a poorer user experience. Additionally, existing simple quota mechanisms are inflexible and fail to accommodate personalization, where different users may benefit from varying weight assignments.
- **(C2)** The performance of multi-channel fusion is highly sensitive to weight combinations. Figure 1 demonstrates the performance variations across different weight combinations on two public datasets: Gowalla and Amazon_Books. We implement nine retrieval channels and observe significant fluctuations in precision

 

and recall by adjusting weight combinations, while keeping the retrieved items from each channel constant. On Gowalla and Amazon_Books, random selection of eight weight combinations results in Recall@200 variations of up to 79.7% and 86.7%, respectively. This underscores the critical need for better optimized weights, which we will discuss further in Section 3.

- **(C3)** Traditional gradient-based methods are unsuitable for this task due to the non-differentiable selection process in multi-channel fusion, complicating weight optimization strategies.

In this paper, we formulate the task of multi-channel fusion in retrieval, laying a cornerstone for future research. We conduct comprehensive analysis and validation, introducing methods for effective multi-channel fusion, unlocking its potential in the retrieval stage to enhance personalized recommendations. Our approach consists of two parts. First, we explore assigning globally unified weights, where weight combinations remain consistent for all users, reflecting current industry practices. We model this problem as a black-box optimization task, where the input consists of weight combinations, and the output is the corresponding retrieval performance. We adopt a two-stage exploration method. In the first stage, the Cross Entropy Method [48] iteratively refines the weight distribution to converge on near-optimal solutions. In the second stage, Bayesian Optimization [15] refines this solution by building a probabilistic model to predict retrieval performance, allowing more efficient exploration of the local search space.

In the second part, we shift from assigning globally unified weights to personalized weights, as users exhibit diverse preferences and behaviors. To optimize this personalized merging process and tackle the non-differentiable selection process of multi-channel fusion, we utilize a policy gradient approach from Reinforcement Learning [62, 65]. These methods go beyond conventional heuristics, paving the way for more intelligent, scalable, and adaptive RS, advancing the frontier of personalized recommendations.

In summary, the contributions of this paper are as follows:

- We are the first to define the challenge of multi-channel fusion in retrieval and demonstrate that systematically optimized weight assignments greatly improve personalized recommendations.
- We propose a two-stage optimization strategy using black-box optimization techniques for non-personalized weight assignment, achieving state-of-the-art (SOTA) performance.
- We introduce a policy gradient-based method for personalized merging, enabling more dynamic and tailored recommendations.
- Extensive experiments on three large-scale, real-world datasets validate the superiority of our approach over current baselines. Moreover, we successfully deploy our method in the recommender system at Company X, resulting in a significant improvement in performance and user experience.

## 2 BACKGROUND

**Multi-Stage Cascade Ranking System.** In modern information retrieval systems, multi-stage cascade ranking is commonly employed [60] to balance efficiency and effectiveness, as illustrated in Figure 1. While complex models [41, 42] often deliver higher accuracy, their inefficiency makes online deployment challenging due to latency constraints [40]. In contrast, simpler models [28, 44] are less powerful but can efficiently process a large number of items

because of their low time complexity. Typically, the system consists of a set of candidate generators and various rankers, structured like a funnel that narrows from bottom to top. Each stage selects the top-$K$ items and passes them to the next. On the left side of Figure 1, we show the approximate output size for each stage.

**Retrieval Strategy.** Retrieval strategies operate as high-level frameworks and can be classified into (1) non-personalized and (2) personalized retrieval. A common non-personalized strategy is promoting popular items, following the 'wisdom of the crowd' [57]. Personalized strategies include U2I and I2I, where U2I links the target user with items they might like directly, while I2I finds items similar to those the user has interacted with. Each strategy provides a distinct approach to discovering items of interest for users.

**Multi-Channel Retrieval.** Multi-channel retrieval [27, 37] is widely adopted in RS, employing independent candidate generators to retrieve distinct item subsets separately [11, 19]. These candidate generators are diverse, utilizing techniques such as associative rules and neural networks, with common methods including matrix factorization [30] and two-tower architectures [64]. As illustrated in Figure 1, the retrieved item subsets are combined to create a comprehensive candidate pool for subsequent ranking stage. The main objective is to expand coverage of users' diverse interests and improve recall rates through various retrieval methods [66], capturing a broad range of user preferences and enhancing performance.

## 3 PRELIMINARIES

### 3.1 Problem Formulation

In this section, we formulate the problem and introduce key notations. Given multiple ranked lists generated by different retrieval channels for each user, the goal is to merge these lists into a unified recommendation set. Let $\mathcal{U}$ and $\mathcal{I}$ denote the sets of users and items, and $K$ represent the total number of retrieval channels. Each channel $k$ provides a ranked list $\mathcal{L}_{uk}$ for user $u \in \mathcal{U}$, where $\mathcal{L}_{uk} \subseteq \mathcal{I}$. The objective is to construct the final recommendation set $\mathcal{R}_u$ for each user by selecting top-ranked items from these lists based on a set of weights, with $|\mathcal{R}_u| = L$, representing a fixed number of items delivered to the subsequent ranking stage. We summarize the notations in Table 4 in Appendix A. We will now detail the merging strategies, constraints, and optimization objectives.

**Merging Strategies:** Merging can be either non-personalized (globally unified) or personalized, depending on whether the weights assigned to each retrieval channel are the same for all users or individualized for each user. In the non-personalized case, each retrieval channel $k$ is assigned a global weight $w_k$. For each channel, we select the top nearest_int($w_k \times L$) items from $\mathcal{L}_{uk}$, forming subsets $\mathcal{L}_{uk}^{(w_k)}$. The final recommendation set $\mathcal{R}_u$ for user $u$ is the union of these selected subsets from all $K$ channels, ensuring no duplicate items, as shown in Equation (1.1).

$$\mathcal{R}_u = \bigcup_{k=1}^{K} \mathcal{L}_{uk}^{(w_k)} \quad \textbf{(1.1)}; \quad \mathcal{R}_u = \bigcup_{k=1}^{K} \mathcal{L}_{uk}^{(w_{uk})} \quad \textbf{(1.2)} \quad (1)$$

In the personalized case, weights $w_{uk}$ vary by user, allowing for a more customized retrieval process. The top nearest_int($w_{uk} \times L$)

items from each list $\mathcal{L}_{uk}$ are selected to form $\mathcal{L}_{uk}^{(w_{uk})}$, and the final recommendation set $\mathcal{R}_u$ is given by Equation (1.2).

**Constraints: (1) Weight Normalization:** Weights $w_k$ for each channel must satisfy Equation (2.1) in the non-personalized case; weights $w_{uk}$ for each user $u$ must satisfy Equation (2.2) in the personalized case. **(2) Weight Bounds:** Weights also have bounds as shown in Equation (3), ensuring no channel is over- or under-represented, reflecting practical requirements in specific scenarios.

$$\sum_{k=1}^{K} w_k = 1 \quad (2.1); \quad \sum_{k=1}^{K} w_{uk} = 1, \quad \forall u \in \mathcal{U} \quad (2.2) \tag{2}$$

$$0 \leq w_{\min} \leq w_k \leq w_{\max} \leq 1 \tag{3}$$

**Optimization Objectives:** To optimize the weights $w_k$ or $w_{uk}$, the goal is to maximize the average evaluation metric across all users, where $\mathcal{T}_u$ is the ground truth set of relevant items for user $u$, and $\text{Eval}(\mathcal{R}_u, \mathcal{T}_u)$ represents the evaluation metric.

$$\max_{\mathbf{w}} \quad \frac{1}{N} \sum_{u \in \mathcal{U}} \text{Eval}(\mathcal{R}_u, \mathcal{T}_u) \tag{4}$$

## 3.2 Rationale Behind Weight Assignment

Figure 1 illustrates how different weight combinations can significantly impact performance of multi-channel fusion. We now explore the rationale for assigning varying weights to the retrieved subsets from different candidate generators. Figure 2 shows our findings on the diversity of candidate generators from multiple perspectives. We implement nine retrieval channels on the Amazon_Books dataset, including associative rule-based methods such as Pop, ItemKNN [51], UserKNN [46], and neural network-based methods like BPR [45], NeuMF [24], SimpleX [36], and LightGCN [23] (detailed in Appendix C). U2I and I2I retrieval strategies are applied for both SimpleX and LightGCN. Each candidate generator retrieves 200 items per user. For items, we measure the pairwise Jaccard similarity [38] between channels, averaged across users:

$$\text{Jaccard}(k_1, k_2) = \frac{1}{N} \sum_{u \in \mathcal{U}} \frac{|\mathcal{L}_{uk_1} \cap \mathcal{L}_{uk_2}|}{|\mathcal{L}_{uk_1} \cup \mathcal{L}_{uk_2}|}, \tag{5}$$

where $|\mathcal{L}_{uk_1} \cap \mathcal{L}_{uk_2}|$ is the number of common items, and $|\mathcal{L}_{uk_1} \cup \mathcal{L}_{uk_2}|$ represents the total unique items. A lower Jaccard score indicates higher diversity across channels. For users, we rank them for each channel based on recall scores, forming a user ranking list $\mathcal{U}_k$. Rank-Biased Overlap (RBO) similarity [61] between the user rankings from two retrieval channels $k_1$ and $k_2$ is formulated as:

$$\text{RBO}(k_1, k_2, p) = (1 - p) \sum_{d=1}^{L} p^{d-1} \frac{|\mathcal{U}_{k_1}^{(d)} \cap \mathcal{U}_{k_2}^{(d)}|}{d}, \tag{6}$$

where $p$ is the persistence parameter ($p$=0.9), controlling emphasis on top-ranked users, and $|\mathcal{U}_{k_1}^{(d)} \cap \mathcal{U}_{k_2}^{(d)}|$ represents the overlap of users at depth $d$. RBO ranges from 0 (no overlap) to 1 (identical rankings). By computing RBO for all channel pairs, we can evaluate how similarly each channel ranks users. Figure 2 visualizes Jaccard and RBO similarity matrices, where most channels exhibit low overlap, indicating **(1) effective multi-channel fusion is crucial** as no single channel covers all user interests, and **(2) personalized weight assignment is necessary** since different channels perform well for different users, supporting argument in Section 1.

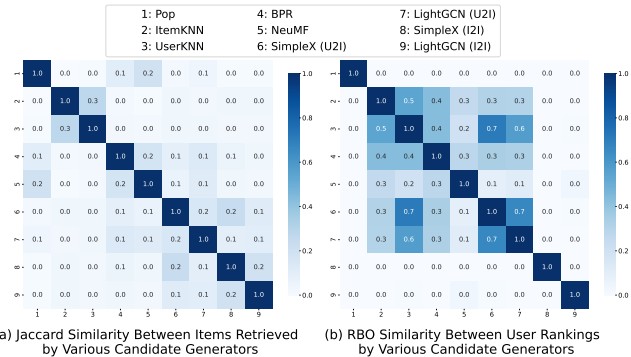

(a) Jaccard Similarity Between Items Retrieved by Various Candidate Generators

(b) RBO Similarity Between User Rankings by Various Candidate Generators

**Figure 2: Diversity among various candidate generators from both item and user perspectives on Amazon_Books.**

## 4 METHODOLOGY

In this section, we explore effective multi-channel fusion strategies in retrieval, starting with globally unified methods, followed by personalized approaches. We present our main idea in Figure 3.

## 4.1 Globally Unified Weight Assignment

In the non-personalized case, the challenge lies in determining the optimal weights for each retrieval channel to maximize the overall performance. Since the objective function, such as overall recall, lacks an explicit mathematical form describing how the weights influence the results, this makes it well-suited for black-box optimization, where the objective is evaluated based on sampled weights without requiring gradient information or predefined problem structure. We adopt a two-phase optimization strategy, which ensures both a broad exploration of the solution space and a more targeted fine-tuning of the best-performing weights. We provide a detailed pseudocode of the training process in Appendix B.

*4.1.1 Cross Entropy Method.* In the first phase, we apply the Cross Entropy Method (CEM), a stochastic optimization technique, to explore the global weight space. Originally introduced by Rubinstein [49] for rare-event probability estimation, CEM uses Kullback-Leibler divergence to update the sampling distribution. It was later adapted for optimization [48, 50], with the search for optimal solutions treated as a rare-event estimation task. CEM iteratively refines the distribution to increase the likelihood of generating near-optimal solutions. Since the weights of various retrieval channels must sum to one, we model the weight vector using the Dirichlet distribution [1]. CEM operates in iterative steps, as outlined below:

**Initialization and Sampling:** We initialize the Dirichlet distribution with parameters $\boldsymbol{\alpha}^{(0)} = [\alpha_1, \alpha_2, \ldots, \alpha_K]^\top$, where each $\alpha_k$ represents the concentration of weight for retrieval channel $k$. The Dirichlet distribution enforces the constraint that weights sum to one. In each iteration, we sample $Q$ weight vectors $\mathbf{w}_1, \mathbf{w}_2, \ldots, \mathbf{w}_Q$ from the current Dirichlet distribution:

$$\mathbf{w}_1, \mathbf{w}_2, \ldots, \mathbf{w}_Q \sim \text{Dirichlet}(\boldsymbol{\alpha}^{(t)}) \tag{7}$$

The probability density function (PDF) of the Dirichlet distribution for a vector $\mathbf{w} = [w_1, w_2, \ldots, w_K]^\top$, with the concentration

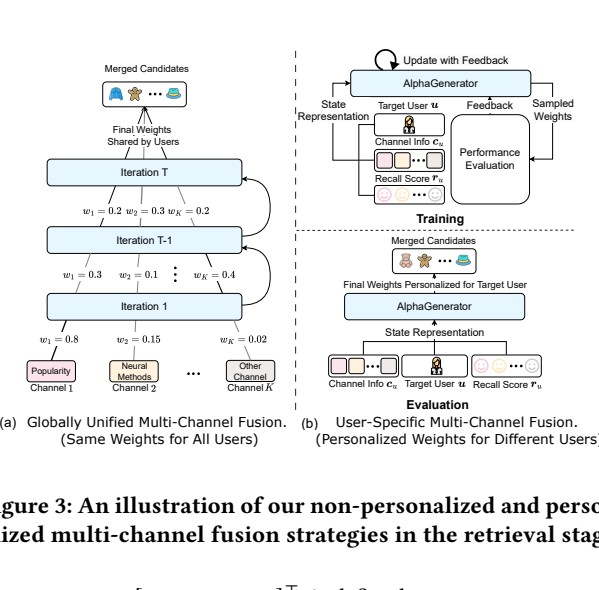

(a) Globally Unified Multi-Channel Fusion. (Same Weights for All Users)

(b) User-Specific Multi-Channel Fusion. (Personalized Weights for Different Users)

**Figure 3: An illustration of our non-personalized and personalized multi-channel fusion strategies in the retrieval stage.**

parameter $\boldsymbol{\alpha} = [\alpha_1, \alpha_2, \ldots, \alpha_K]^\top$, is defined as:

$$f(\mathbf{w}; \boldsymbol{\alpha}) = \frac{\Gamma\left(\sum_{i=1}^{K} \alpha_i\right)}{\prod_{i=1}^{K} \Gamma(\alpha_i)} \prod_{i=1}^{K} w_i^{\alpha_i - 1} \tag{8}$$

where $\Gamma(\cdot)$ is the Gamma function. These samples represent different possible weight combinations across the retrieval channels.

**Performance Evaluation:** For each sampled weight vector $\mathbf{w}_i$, we compute the retrieval performance $S(\mathbf{w}_i)$ using a metric like expected recall. This metric acts as a proxy for how well the weights enhance retrieval results. The performance is evaluated for all samples without requiring an explicit objective function.

**Selecting Elite Samples:** After evaluating all $Q$ samples, we rank them in descending order and select the top $q$-percentile as the elite set. The performance threshold $\hat{\gamma}_t$ is the score of the lowest-ranked sample in the elite set, where $Q^e = \lceil qQ \rceil$ is the number of elite samples. All samples with $S(\mathbf{w_i}) \geq \hat{\gamma}_t$ are retained.

$$\hat{\gamma}_t = S_{(Q - Q^e + 1)} \tag{9}$$

**Parameter Update (Cross-Entropy Step):** We iteratively refine the weight distribution to focus on better-performing solutions by updating $\boldsymbol{\alpha}$ at each iteration. In Equation (10), the new parameters $\boldsymbol{\alpha}^*$ maximize the likelihood of generating these elite samples, where $\mathbb{I}(S(\mathbf{w}_i) \geq \hat{\gamma}_t)$ is an indicator function that selects the elite samples.

$$\boldsymbol{\alpha}^* = \arg\max_{\boldsymbol{\alpha}} \frac{1}{Q} \sum_{j=1}^{Q} \mathbb{I}(S(\mathbf{w}_j) \geq \hat{\gamma}_t) \log f(\mathbf{w}_j; \boldsymbol{\alpha}) \tag{10}$$

Once $\boldsymbol{\alpha}^*$ is found, the parameters are smoothly updated using a learning rate $\eta_1$. This weighted average gradually shifts the distribution toward elite samples while maintaining stability.

$$\boldsymbol{\alpha}^{(t+1)} = (1 - \eta_1) \cdot \boldsymbol{\alpha}^{(t)} + \eta_1 \cdot \boldsymbol{\alpha}^* \tag{11}$$

*4.1.2 Bayesian Optimization.* After the global exploration with CEM, we refine the solution using Bayesian Optimization (BayesOpt), which fine-tunes the Dirichlet distribution's parameters in a constrained search space. Specifically, the search space for parameter $\boldsymbol{\beta}$ is set to the range $[0.5\boldsymbol{\alpha}^{(t)}, 1.5\boldsymbol{\alpha}^{(t)}]$, where $\boldsymbol{\alpha}^{(t)}$ is the result from

the CEM stage, ensuring that the optimization remains focused on promising regions. BayesOpt has two key components [15, 52]:

**Surrogate Model:** A Gaussian Process (GP) models the objective function $S(\cdot)$, such as the expected recall. The GP provides both predictions and uncertainty estimates for unexplored regions:

$$S(\boldsymbol{\beta}) \sim \mathcal{GP}(\mu(\boldsymbol{\beta}), k(\boldsymbol{\beta}, \boldsymbol{\beta}')) \tag{12}$$

where $\mu(\cdot$ is the predicted mean, and $k(\cdot)$ is the covariance function.

**Acquisition Function:** This function selects the next sample by balancing exploration and exploitation. The next Dirichlet parameters are chosen to maximize expected improvement (EI) in retrieval performance, with $S_{\text{best}}$ representing the best performance so far.

$$\arg\max_{\boldsymbol{\beta}} \mathbb{E}[\max(S(\boldsymbol{\beta}) - S_{\text{best}}, 0)] \tag{13}$$

The process iteratively refines $\boldsymbol{\beta}$ to converge toward optimal parameters. Once $\boldsymbol{\beta}$ is determined, the final step is to derive the optimal weight vector $\mathbf{w}$. Since $\boldsymbol{\beta}$ parameterizes a Dirichlet distribution, the optimal weights are the expected value of the distribution.

$$\mathbb{E}[\mathbf{w}] = \frac{\boldsymbol{\beta}}{\sum_i \beta_i} \tag{14}$$

## 4.2 Personalized Weight Assignment

Globally unified weights provide a general solution but overlook individual user preferences. Personalized fusion are essential, as users benefit from different retrieval combinations based on their unique behaviors and preferences. Due to the non-differentiable nature of the selection process in multi-channel fusion, traditional gradient-based methods like SGD are unsuitable. To address this, we employ a policy gradient approach (PG) from Reinforcement Learning [62, 65] to optimize the merging strategy. We model the weight assignment as a policy that generates a probability distribution over possible weights for each user. This policy, parameterized by a neural network, takes as input the user representation $\boldsymbol{u}$, the recall scores from each retrieval channel $\boldsymbol{r}_u = [r_{u1}, r_{u2}, \ldots, r_{uK}]^\top$, and the retrieval channel representations $\{\boldsymbol{c}_{uk}\}_{k=1}^{K}$. These components together constitute the state $s_u$ for user $u$:

$$s_u = \left(\boldsymbol{u}, \boldsymbol{r}_u, \{\boldsymbol{c}_{uk}\}_{k=1}^{K}\right) \tag{15}$$

Our policy outputs the parameters $\boldsymbol{\alpha}_u$ of a Dirichlet distribution for each user $u$, which determines the weight distribution $\mathbf{w}_u$.

*4.2.1 Model Architecture.* During forward propagation, we compute Dirichlet parameters $\boldsymbol{\alpha}_u$ for each user, which are then used to sample weights $\mathbf{w}_u$ for merging retrieval results. Let $\boldsymbol{u} \in \mathbb{R}^d$, $\boldsymbol{c}_{uk} \in \mathbb{R}^d$, $r_u \in \mathbb{R}^K$, and $h$ represent the hidden dimension size. After training the single-channel models, $r_u$ remains a fixed constant. In our method, $\boldsymbol{u}$ is the user representation generated from one of the pre-trained retrieval models, and $\boldsymbol{c}_{uk}$ is obtained by pooling the top-$m$ item representations from channel $k$ of the same model. Since the retrieval results vary for each user, the channel representations $\boldsymbol{c}_{uk}$ are user-dependent. First, we apply linear transformations followed by ReLU activations to both the user representations and each channel's representations, where $\boldsymbol{h}_u \in \mathbb{R}^h$, $\boldsymbol{h}_{c_{uk}} \in \mathbb{R}^h$.

$$\boldsymbol{h}_u = \text{ReLU}(\mathbf{W}_u \boldsymbol{u} + \boldsymbol{b}_u), \quad \boldsymbol{h}_{c_{uk}} = \text{ReLU}(\mathbf{W}_c \boldsymbol{c}_{uk} + \boldsymbol{b}_c) \tag{16}$$

Here, $\mathbf{W}_u \in \mathbb{R}^{h \times d}, \boldsymbol{b}_u \in \mathbb{R}^h, \mathbf{W}_c \in \mathbb{R}^{h \times d}$ and $\boldsymbol{b}_c \in \mathbb{R}^h$, are learnable parameters. Next, we compute the dot product between the transformed user and channel representations to model user preference toward each channel $v_{uk} \in \mathbb{R}$:

$$v_{uk} = \boldsymbol{h}_u^\top \boldsymbol{h}_{c_{uk}}, \quad \boldsymbol{v}_u = [v_{u1}, v_{u2}, \ldots, v_{uK}]^\top \in \mathbb{R}^K \quad (17)$$

We combine attention scores with user recall scores from each channel to generate combined scores $\boldsymbol{e}_u \in \mathbb{R}^K$. $\boldsymbol{\alpha}_u \in \mathbb{R}^K$ are computed using a scaled hyperbolic tangent activation, with $\delta_{\max}$ controlling the maximum adjustment. To ensure $\boldsymbol{\alpha}_u$ remains positive, we apply a ReLU activation and add a small constant $\epsilon$ to avoid zero values. This entire process of generating $\boldsymbol{\alpha}_u$ from state $s_u$ in Equation (15) is referred to as AlphaGenerator, as shown in Figure 3.

$$\boldsymbol{e}_u = \boldsymbol{v}_u + \boldsymbol{r}_u, \quad \boldsymbol{\alpha}_u = \delta_{\max} \cdot \tanh(\boldsymbol{e}_u), \quad \boldsymbol{\alpha}_u = \text{ReLU}(\boldsymbol{\alpha}_u) + \epsilon \quad (18)$$

After computing $\boldsymbol{\alpha}_u$, we sample the weight vector $\mathbf{w}_u \in \mathbb{R}^K$ from the Dirichlet distribution. These weights merge the retrieval results across channels for user $u$, and the reward $R(s_u, \mathbf{w}_u)$ is calculated based on a performance metric of the merged results. During evaluation, we use Equation (14) to compute the expected value of the distribution, which serves as the final optimal weights $\mathbf{w}_u$.

*4.2.2 Objective Function.* Our objective is to maximize the expected reward $J(\theta)$, where $\theta$ represents the parameters of the neural network and $R(s_u, \mathbf{w}_u)$ is the reward obtained by applying weights $\mathbf{w}_u$ in state $s_u$. The policy $\pi_\theta(\mathbf{w}_u|s_u)$ is defined as a Dirichlet distribution parameterized by $\boldsymbol{\alpha}_u$, where $\boldsymbol{\alpha}_u$ is computed from the neural network named AlphaGenerator based on the state $s_u$.

$$J(\theta) = \frac{1}{N} \sum_{u \in \mathcal{U}} \mathbb{E}_{\mathbf{w}_u \sim \pi_\theta(\mathbf{w}_u|s_u)} [R(s_u, \mathbf{w}_u)] \quad (19)$$

$$\pi_\theta(\mathbf{w}_u|s_u) = \text{Dirichlet}(\boldsymbol{\alpha}_u) \quad (20)$$

$$\boldsymbol{\alpha}_u = f_\theta(s_u) = f_\theta\left(\boldsymbol{u}, \boldsymbol{r}_u, \{\boldsymbol{c}_{uk}\}_{k=1}^K\right) \quad (21)$$

To maximize $J(\theta)$, we compute the gradient with respect to $\theta$, as in Equation (22), using Monte Carlo sampling. For each user $u$, we sample $S$ weight vectors $\{\mathbf{w}_{u,i}\}_{i=1}^S$ from the policy $\pi_\theta(\mathbf{w}_u|s_u)$ and compute the corresponding rewards $\{R_{u,i}\}_{i=1}^S$.

$$\nabla_\theta J(\theta) = \nabla_\theta \left(\frac{1}{N} \sum_{u \in \mathcal{U}} \mathbb{E}_{\mathbf{w}_u \sim \pi_\theta(\mathbf{w}_u|s_u)} [R(s_u, \mathbf{w}_u)]\right)$$

$$= \frac{1}{N} \sum_{u \in \mathcal{U}} \nabla_\theta \mathbb{E}_{\mathbf{w}_u \sim \pi_\theta(\mathbf{w}_u|s_u)} [R(s_u, \mathbf{w}_u)] \quad (22)$$

$$= \frac{1}{N} \sum_{u \in \mathcal{U}} \mathbb{E}_{\mathbf{w}_u \sim \pi_\theta(\mathbf{w}_u|s_u)} [R(s_u, \mathbf{w}_u) \nabla_\theta \log \pi_\theta(\mathbf{w}_u|s_u)]$$

$$\approx \frac{1}{N} \sum_{u \in \mathcal{U}} \left(\frac{1}{S} \sum_{i=1}^S R_{u,i} \nabla_\theta \log \pi_\theta(\mathbf{w}_{u,i}|s_u)\right)$$

We define the loss function as the negative expected reward over all users to perform gradient ascent on $J(\theta)$:

$$L(\theta) = -J(\theta) \quad (23)$$

To prevent overfitting and encourage the learned weights to stay close to the global weights $\mathbf{w}_{\text{global}}$ from Section 4.1, we add a regularization term that penalizes large deviations, with $\lambda$ controlling

the penalty strength. The total loss function is a combination of the policy loss and the regularization term.

$$L_{\text{reg}}(\theta) = \lambda \frac{1}{N} \sum_{u \in \mathcal{U}} \left(\frac{1}{S} \sum_{i=1}^S \left\|\mathbf{w}_{u,i} - \mathbf{w}_{\text{global}}\right\|_2^2\right) \quad (24)$$

$$L_{\text{total}}(\theta) = L(\theta) + L_{\text{reg}}(\theta) \quad (25)$$

## 5 EXPERIMENTS

In this section, we detail our experimental settings and results on three large-scale public datasets. We evaluate our methods, including the Cross Entropy Method (CEM), Bayesian Optimization (BayesOpt), and the policy gradient approach (PG), against strong baseline models, demonstrating state-of-the-art performance.

### 5.1 Dataset and Experimental Flow

We use three real-world datasets: Gowalla[1], Amazon_Books[2], and Tmall[3]. Dataset statistics are shown in Table 1. Only users with at least 10 recorded behaviors are included [69]. We split the datasets into training, validation, and test sets in a 5:2:3 ratio based on timestamps. For each implicit feedback instance, we randomly select 100 negative samples for Gowalla and Amazon_Books, and 200 negative samples for Tmall. Further details on the datasets and implementation details can be found in Appendix C.1 and C.3.

### 5.2 Baselines and Evaluation Metrics

For each dataset, we implement nine retrieval channels, including associative rule-based methods such as Pop, ItemKNN [51], UserKNN [46], and neural network-based methods like BPR [45], NeuMF [24], SimpleX [36], and LightGCN [23]. For SimpleX and LightGCN, we apply both U2I and I2I retrieval strategies to retrieve distinct item subsets, enhancing diversity (see Appendix C.2 for details). Additionally, we implement two basic merging methods: the first is equal-weight merging, where all retrieval channels are assigned the same weight; the second is statistical merging, where weights are normalized based on the proportion of retrieved items clicked by users. In statistical merging, channels with higher performance are usually assigned greater weights. It simulates heuristic weighting methods commonly used in industry practices.

To evaluate the effectiveness of different methods, we use Precision@L (P@L), Recall@L (R@L), and F-Measure@L (F1@L) metrics [32], as our focus is on the number of relevant items returned rather than specific ranking order. Using the notations in Table 4, we present the formulas for these metrics in Appendix C.5.

**Table 1: Statistics of the experimental datasets.**

| Dataset | # Users | # Items | # Interactions | Sparsity |
|---|---|---|---|---|
| Gowalla | 68,709 | 1,247,158 | 3,831,386 | 99.99% |
| Amazon_Books | 294,739 | 1,477,922 | 8,654,619 | 99.99% |
| Tmall | 385,359 | 2,184,385 | 34,255,087 | 99.99% |

---

[1]https://snap.stanford.edu/data/loc-gowalla.html.
[2]https://jmcauley.ucsd.edu/data/amazon/amazonbooks.
[3]https://tianchi.aliyun.com/dataset/53.

**Table 2: Overall performance of various methods on three public datasets. The top two results in each column are highlighted to indicate SOTA performance. The strongest baseline is underlined, and relative improvement (RelImp) is reported.**

| Model | Gowalla | | | Amazon_Books | | | Tmall | | |
|---|---|---|---|---|---|---|---|---|---|
| | P@200 | R@200 | F1@200 | P@200 | R@200 | F1@200 | P@200 | R@200 | F1@200 |
| Candidate Generators | | | | | | | | | |
| Pop | 0.25% | 3.48% | 0.42% | 0.08% | 2.44% | 0.14% | 0.18% | 3.29% | 0.31% |
| ItemKNN [51] | 0.24% | 3.39% | 0.43% | 0.10% | 2.70% | 0.18% | 0.24% | 2.91% | 0.38% |
| UserKNN [46] | 0.64% | 9.18% | 1.08% | 0.10% | 2.01% | 0.18% | 0.22% | 4.63% | 0.38% |
| BPR [45] | 0.56% | 7.28% | 0.95% | 0.16% | 4.89% | 0.30% | 0.32% | 5.66% | 0.55% |
| NeuMF [24] | 0.68% | 9.32% | 1.16% | 0.17% | 4.74% | 0.31% | 0.38% | 6.63% | 0.64% |
| SimpleX (U2I) [36] | 0.74% | 11.54% | 1.30% | 0.25% | 7.94% | 0.46% | 0.50% | 8.72% | 0.85% |
| SimpleX (I2I) [36] | 0.62% | 10.22% | 1.09% | 0.19% | 6.47% | 0.36% | 0.30% | 5.98% | 0.51% |
| LightGCN (U2I) [23] | 0.74% | 11.82% | 1.32% | 0.27% | 8.29% | 0.50% | 0.40% | 6.84% | 0.68% |
| LightGCN (I2I) [23] | 0.65% | 11.58% | 1.16% | 0.18% | 6.00% | 0.33% | 0.31% | 6.32% | 0.53% |
| Basic Merging Methods | | | | | | | | | |
| Equal-Weight Merging | 0.71% | 11.26% | 1.24% | 0.23% | 7.75% | 0.44% | 0.49% | 9.17% | 0.84% |
| Statistical Merging | 0.79% | 12.45% | 1.38% | 0.25% | 8.34% | 0.47% | 0.51% | 9.43% | 0.88% |
| Our Methods | | | | | | | | | |
| CEM (non-personalized) | 0.82% | 13.08% | 1.43% | 0.27% | 8.77% | 0.50% | 0.53% | 9.65% | 0.90% |
| BayesOpt (non-personalized) | **0.82%(2)** | **13.22%(2)** | **1.44%(2)** | **0.27%(2)** | **8.85%(2)** | **0.51%(2)** | **0.53%(2)** | **9.68%(2)** | **0.90%(2)** |
| PG (personalized) | **0.85%(1)** | **13.58%(1)** | **1.49%(1)** | **0.29%(1)** | **9.21%(1)** | **0.53%(1)** | **0.55%(1)** | **10.02%(1)** | **0.93%(1)** |
| RelImp (non-personalized) | 3.79%↑ | 6.18%↑ | 4.35%↑ | 0.00%↑ | 6.12%↑ | 2.00%↑ | 3.92%↑ | 2.65%↑ | 2.27%↑ |
| RelImp (personalized) | 7.59%↑ | 9.08%↑ | 7.97%↑ | 7.41%↑ | 10.43%↑ | 6.00%↑ | 7.84%↑ | 6.26%↑ | 5.68%↑ |

## 5.3 Performance Comparison

Table 2 presents the recommendation performance of different models in terms of precision, recall, and F-measure across three datasets, from which we have the following observations:

- All three methods we propose significantly outperform existing baselines. Specifically, BayesOpt (non-personalized) and PG (personalized) improve upon the strongest baseline by 6.18% and 9.08% in R@200 on Gowalla and 6.12% and 10.43% on Amazon_Books. These results underscore two key contributions: (1) our method offers a more effective merging strategy, and (2) personalized multi-channel fusion further enhances performance over the industry-standard globally unified weighting approach, emphasizing the importance of personalization.
- Compared to rule-based methods like Pop, neural network-based approaches, particularly state-of-the-art models such as SimpleX [36] and LightGCN [23], demonstrate superior performance.
- Even simple multi-channel merging methods, such as statistical merging with heuristic-based weight assignments, easily surpass the performance of the best single-channel retrieval models, demonstrating the effectiveness of multi-channel retrieval.

## 5.4 In Depth Analysis

*5.4.1 Context Dependency.* Figure 4 presents the global weights **w** optimized through BayesOpt. As shown, SimpleX (U2I), SimpleX

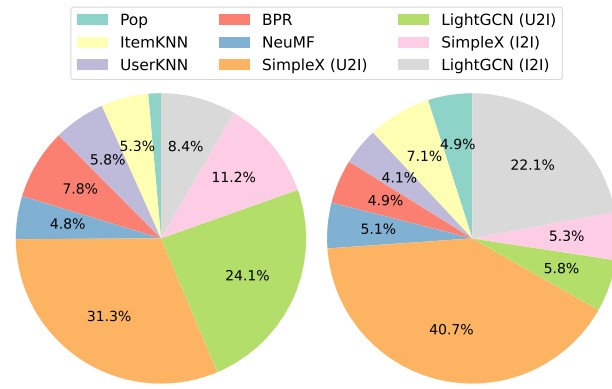

(a) Optimal Weights on Amazon Books   (b) Optimal Weights on Tmall

**Figure 4: Optimal weights for various retrieval channels generated by Bayesian Optimization on Amazon Books and Tmall. Proportions below 2% are omitted for clarity.**

(I2I), LightGCN (U2I), and LightGCN (I2I) account for the largest proportions, while the remaining five models contribute relatively less. Furthermore, we observe that even with the same nine retrieval models, the optimal weights vary across different datasets and

scenarios. This highlights that the distribution of weights in multi-channel retrieval is highly context-dependent, with no fixed rule dictating how much weight each model should carry.

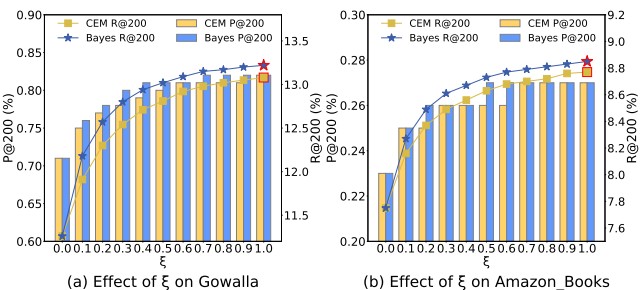

(a) Effect of $\xi$ on Gowalla    (b) Effect of $\xi$ on Amazon_Books

**Figure 5: Effect of $\xi$ on Gowalla and Amazon_Books.**

*5.4.2  Globally Unified Weight Assignment.* To further investigate the distribution parameters $\boldsymbol{\alpha}$ optimized by CEM and BayesOpt, we introduce the following adjustment in Equation (26):

$$\boldsymbol{\alpha} = \xi \cdot \boldsymbol{\alpha}^{(0)} + (1 - \xi) \cdot \boldsymbol{\alpha}^{(t)}, \tag{26}$$

where $\boldsymbol{\alpha}^{(t)}$ represents the optimal distribution parameters obtained from CEM or BayesOpt, and $\xi$ varies from 0 to 1 with intervals of 0.1. Figure 5 illustrates how retrieval performance changes as $\xi$ increases. We observe a steady improvement in performance as $\xi$ grows, indicating that the global weights optimized by our methods are well-founded and not a result of random chance.

*5.4.3  Personalized Weight Assignment.* We now provide a detailed analysis of our personalized merging strategy PG.

**Hyperparameter Study.** Regularization weight $\lambda$ in Equation (24) is a key hyperparameter in the PG method. Figure 6 shows the impact of different $\lambda$ values on PG performance. When $\lambda$ is too small, the constraint between personalized and global weights is too weak, resulting in suboptimal performance. Increasing $\lambda$ initially improves results, but when it becomes too large, performance declines as the tight constraint limits the potential of personalization.

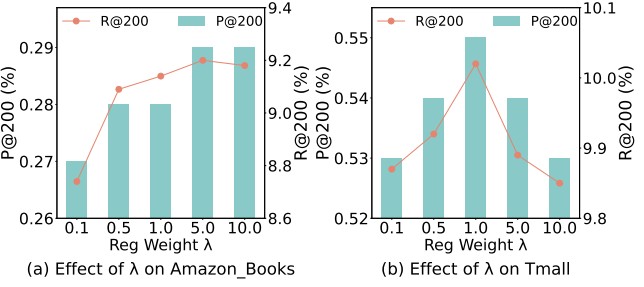

(a) Effect of $\lambda$ on Amazon_Books    (b) Effect of $\lambda$ on Tmall

**Figure 6: Effect of regularization weight $\lambda$ on Amazon_Books and Tmall.**

**Visualization.** We randomly select 2,000 users on Amazon_Books and Tmall to visualize the personalized weight distributions generated by PG in Figure 7, which reveal several key insights:

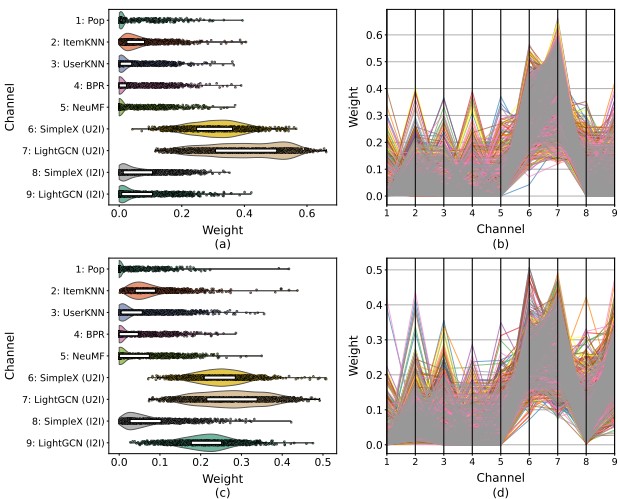

**Figure 7: Visualization of personalized weights across Amazon_Books and Tmall. (a) (b) Channel weight distribution and patterns on Amazon_Books (violin plot and parallel coordinate plot). (c) (d) Channel weight distribution and patterns on Tmall (violin plot and parallel coordinate plot).**

- **Weight Distribution Consistency**: SimpleX and LightGCN have the largest weights across both datasets, which is consistent with the global weight assignment results.
- **User-Specific Diversity:** The parallel coordinate plots highlight diverse weight distributions across users, with multiple peaks indicating varying user preferences for different channels.
- **Performance-Weight Relationship:** The weight assigned to a retrieval channel is not strictly tied to its performance. For instance, despite ItemKNN's lower retrieval performance, its weight remains notable, suggesting that factors such as item overlap between channels also play a role in weight optimization.

See Appendix D for further discussion of the experiments.

## 6  REAL WORLD DEPLOYMENT

To validate the effectiveness of our multi-channel fusion strategies in real-world scenarios, we deploy our method in one main recommendation scenario (called 'Smart Living') at Company X, a main-stream bank company. This application serves millions of daily active users, generating billions of user logs through implicit feedback, such as click behavior. For further details on the deployment process, please refer to the discussion in Appendix E.

### 6.1  Offline Evaluation

For the offline experiment, we use a daily updated dataset collected from July 2024 to August 2024 in the 'Smart Living' recommendation scenario for training and evaluation. The scenario involves 11 retrieval channels. Under real-world conditions, the number of items retrieved by each channel may vary; for instance, cold-start users with no interaction history may yield insufficient results from the I2I retrieval method. To address this, we pad the shorter retrieval channels to match the longest one, aligning with our problem formulation in Section 3. The dataset includes true exposure data, capturing items where users paused briefly instead of scrolling past.

Since users typically engage with only a few to a few dozen items during a recommendation session, we evaluate the top 10 items using P@10, R@10, and F1@10. As shown in Table 3, CEM outperforms the current production strategy significantly, delivering approximately a 28.6% improvement in offline metrics.

**Table 3: Comparison of different merging strategies on real-world recommendation scenarios.**

| Strategy | P@10 | R@10 | F1@10 | CTR |
|---|---|---|---|---|
| Equal-Weight Merging | 4.81% | 17.20% | 7.52% | / |
| Current Production Strategy | 8.09% | 28.95% | 12.65% | 1.77% |
| CEM (globally unified) | **10.41%** | **37.22%** | **16.27%** | **2.07%** |

## 6.2 Online Evaluation

Besides offline experiments, we conduct a five-day online A/B test in October 2024, deploying our method in the 'Smart Living' recommendation scenario of Company X. As mentioned earlier, industry recommender systems typically enforce bounds on multi-channel weight assignments to ensure balanced representation across channels, as shown in Equation 3. This makes equal-weight merging infeasible in real-world deployments. Additionally, due to the current pipeline and engineering limitations, we could not implement personalized multi-channel fusion methods like our PG approach. Instead, we deploy our globally unified weight assignment strategy CEM, which aligns with common industry practice.

The control group uses the heuristic-based merging strategy from the current production system, while the test group implements our globally unified CEM strategy at the retrieval stage. Both groups use the same ranking strategy to ensure a fair comparison. We evaluate performance using Click-Through Rate (CTR), defined as: $CTR = \frac{\#clicks}{\#impressions}$ where #clicks and #impressions are the number of clicks and impressions. We report the average results in Table 3, and Figure 8 presents the daily and hourly improvements of CEM over the current strategy. It is evident that CEM significantly outperforms the baseline with a CTR increase of 12% to 20% (average 17%), highlighting the critical role of our optimized multi-channel fusion in recommendation performance.

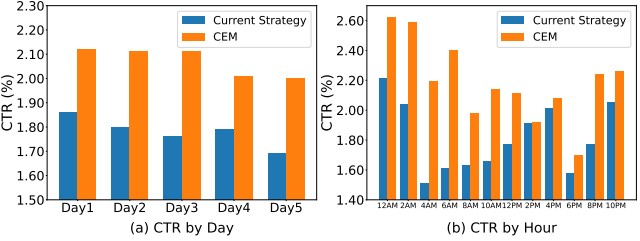

(a) CTR by Day        (b) CTR by Hour

**Figure 8: Daily and hourly CTR results from online A/B test in the 'Smart Living' scenario.**

## 7 RELATED WORK

**Retrieval Methods in Recommendation.** Retrieval is the process of efficiently selecting relevant item candidates that match user interests, also referred to as candidate generation or matching [11, 26]. Retrieval methods vary widely, which can be broadly categorized into two types: (1) non-personalized and (2) personalized retrieval [27]. Non-personalized retrieval highlights popular items or trending content, which, while not tailored to individual preferences, often attract user clicks due to their widespread appeal. In contrast, personalized retrieval customizes recommendations to align with specific user preferences, significantly boosting engagement and retention. Common examples include user-to-item (U2I) [30] and item-to-item (I2I) [51] retrieval. Diving deeper into model structures, there are shallow structures like neighborhood-based collaborative filtering (CF) approaches such as ItemKNN [51] and UserKNN [46], as well as matrix factorization (MF)-based CF approaches [30]. With the advancement of deep learning, there has been a shift toward more sophisticated architectures, including two-tower retrieval models [11, 19, 28, 36], autoencoder-based models [33, 35, 53, 63], graph embedding-based models [4, 20, 39, 59], graph neural network-based models [5, 23, 56], tree-based models [68, 69, 71], and multi-interest retrieval models [9, 31, 58]. These diverse retrieval channels improve both relevance and diversity of recommendation results. In our experiments, we select models from different categories to minimize overlap and enhance diversity.

**Multi-Channel Retrieval.** Multi-channel retrieval is widely used in modern industry practices for cascade recommender systems. MIC [37] effectively aligns users and items based on semantic similarity across channels (U2U, I2I, U2I), leveraging rich cross-channel information. Hron et al. [25] empirically and theoretically explore the differences between single- and two-stage recommenders, showing that when each candidate generator specializes in a different subset of the item pool, performance improves significantly. Similar concepts include recommender ensembling [7], such as weighted hybrid, cross-harmonic, and meta-model mixed recommendation algorithms [6]. However, none of these approaches offer a scientific or systematic solution for multi-channel fusion in the retrieval stage, which is a critical aspect in real-world implementations.

**Combinatorial Optimization.** In addition to the Cross Entropy Method [50] and Bayesian Optimization [15] we use, other well-known approaches for combinatorial optimization include simulated annealing [2, 10, 12, 47], later extended in [22] and [29], as well as tabu search [17] and genetic algorithms [18]. More recent methods include nested partitioning [54], stochastic comparison [3], and ant colony optimization [13, 21].

## 8 CONCLUSION

In this paper, we address the challenge of multi-channel fusion in retrieval. Moving beyond the heuristic and manual methods commonly used in industry, we demonstrate that our optimized weight combinations significantly enhance personalized recommendations. By leveraging black-box optimization and a policy gradient-based method, we provide a user-tailored approach that advances beyond simple quota mechanisms. Extensive experiments across multiple datasets show our approach consistently outperforms existing baselines, and its successful deployment in real-world systems results in notable improvements in both performance and user satisfaction, offering a scalable solution for multi-channel fusion.

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

## A  NOTATIONS

We summarize the key notations and their corresponding descriptions used in this paper in Table 4.

**Table 4: Notations and descriptions.**

| Notation | Description. |
|---|---|
| $\mathcal{U}, \mathcal{I}$ | Sets of all users and items, respectively. |
| $N, M$ | Total number of users and items, respectively. |
| $K$ | Total number of retrieval channels (candidate generators). |
| $L$ | Fixed number of items sent to the subsequent ranking stage. |
| $\mathcal{L}_{uk}$ | Ranked list of items from retrieval channel $k$ for user $u$. |
| $w_k, w_{uk}$ | Weight assigned to retrieval channel $k$ (for user $u$). |
| $\mathcal{L}_{uk}^{(w_k)}, \mathcal{L}_{uk}^{(w_{uk})}$ | Subset of top-ranked items selected from $\mathcal{L}_{uk}$. |
| $\mathcal{R}_u$ | Final set of recommended items for user $u$. |
| $\mathcal{T}_u$ | Ground truth set of relevant items for user $u$. |
| $\mathcal{U}_k$ | Ranked list of users based on their recall score from retrieval channel $k$. |
| $\mathcal{U}_k^{(d)}$ | Top-$d$ ranked users in channel $k$. |
| $\boldsymbol{\alpha}, \boldsymbol{\beta}$ | Parameters of the Dirichlet distribution. |
| $Q, q$ | Number of samples per iteration in CEM, and the elite fraction selected. |
| $\boldsymbol{u}, \boldsymbol{r}_u, \boldsymbol{c}_k$ | User representation, recall scores from each channel, and channel $k$ representation. |
| $\eta_1, \eta_2$ | Learning rate in CEM and PG, respectively. |
| $\lambda, \xi$ | Regularization weight in PG, and tuning parameter in Equation (26). |
| $\delta_{\max}, \epsilon$ | Maximum adjustment magnitude, and a small positive constant in Equation (18). |
| $S$ | Number of sampled weights for each user in Equation (22). |

## B  PSEUDOCODE FOR TRAINING PROCEDURE OF GLOBALLY UNIFIED MERGING

---

**Algorithm 1** Globally Unified Weight Assignment

---

**Require:** Elite fraction $q$, number of samples per iteration $Q$, performance evaluation function $\mathcal{S}(\cdot)$, acquisition function $a_{\mathrm{EI}}(\cdot)$, number of BayesOpt iterations $T$

1: **Stage 1: Cross Entropy Method (CEM)**
2: **Initialize** concentration parameter vector $\boldsymbol{\alpha}^{(0)}$, $t = 0$
3: **repeat**
4:     Sample $Q$ weight vectors from Dirichlet($\boldsymbol{\alpha}^{(t)}$)
5:     Evaluate retrieval performance for each sampled weight:
$$\mathcal{S}(\mathbf{w}_i) \quad \text{for} \quad i = 1, 2, \ldots, Q$$
6:     Select the elite samples based on performance
7:     Update $\alpha^{(t+1)}$ using elite samples
8:     Increment $t$
9: **until** convergence
10: Set $\beta^{(0)} = \alpha^{(t)}$ // *Initialize BayesOpt with final CEM parameters*
11: **Stage 2: Bayesian Optimization (BayesOpt)**
12: Constrained Search Space: $[0.5\boldsymbol{\beta}^{(0)}, 1.5\boldsymbol{\beta}^{(0)}]$
13: Initialize Gaussian Process (GP) model $\mathcal{GP}$ with $\boldsymbol{\beta}^{(0)}$
14: **for** $t = 1, 2, \ldots, T$ **do**
15:     Fit GP model to observed data
16:     Predict objective function $S(\boldsymbol{\beta})$ for unexplored regions
17:     Compute acquisition function $a_{\mathrm{EI}}(\boldsymbol{\beta})$
18:     Find next sample $\boldsymbol{\beta}^{(t+1)} = \arg\max_{\boldsymbol{\beta}} a_{\mathrm{EI}}(\boldsymbol{\beta})$
19:     Evaluate objective function $S(\boldsymbol{\beta}^{(t+1)})$
20:     Update GP with new data $(\boldsymbol{\beta}^{(t+1)}, S(\boldsymbol{\beta}^{(t+1)}))$
21: **end for**
22: **return** Optimal weight vector $\mathbf{w}$ using Equation (14)

---

## C  EXPERIMENTAL CONFIGURATION

### C.1  Dataset Description

We conduct experiments on three real-world, large-scale datasets.

**Gowalla**[4] dataset is collected from the Gowalla social network, a location-based platform where users could check in at physical locations and share their activities with friends.

**Amazon_Books**[5] dataset is a subset of the Amazon review dataset, which contains millions of reviews written by Amazon customers for various products on the Amazon e-commerce platform.

**Tmall**[6] dataset is provided by Ant Financial Services, containing users' online and on-site behavior from July to November 2015.

### C.2  Baseline Description

In our experiment, we implement nine retrieval channels on each of the three datasets. Brief descriptions of these channels are provided:

- **Pop** is a basic model that consistently recommends the most popular items.
- **ItemKNN [51]** is a simple model that calculates item similarity using the interaction matrix.
- **UserKNN [46]** is a simple model that calculates user similarity using the interaction matrix.
- **BPR [45]** is a basic matrix factorization model trained using a pairwise learning approach.
- **NeuMF [24]** enhances matrix factorization with a neural network by replacing the dot product with an MLP, offering a more precise model of user-item interactions.
- **SimpleX [36]** is a straightforward two-tower retrieval model that stands out for its loss function. It incorporates a larger pool of negative samples and filters out uninformative ones using a threshold. Additionally, it balances the loss between positive and negative samples by applying relative weights.
- **LightGCN [23]** focuses solely on the core aspect of GCN, neighborhood aggregation, for collaborative filtering. It learns user and item embeddings through linear propagation on the user-item interaction graph, and combines the embeddings from all layers using a weighted sum to produce the final embedding.

For SimpleX (I2I) and LightGCN (I2I), we take the user's three most recent interactions from the training set and retrieve the 80 most similar items for each. After merging and removing duplicates, if fewer than 200 items remain, we continue adding more until we reach 200 items. If the final set exceeds 200 items, we truncate it to ensure the retrieved item set contains exactly 200 items.

### C.3  Implementation Details

We implement all methods with PyTorch using Recbole [67], a comprehensive framework for recommendation models. The hyperparameters for the nine retrieval channels are provided in Appendix C.4, with each channel retrieving 200 items. For globally unified weight assignment, we initialize the Dirichlet distribution with $\boldsymbol{\alpha}^{(0)} = [1, 1, \ldots, 1]^{\top}$. The learning rate $\eta_1$ in Equation (11) is set to 0.1 with a decay factor of 0.95 applied if no performance improvement is observed. In each round, 60 samples are drawn,

---

[4]https://snap.stanford.edu/data/loc-gowalla.html.
[5]https://jmcauley.ucsd.edu/data/amazon/amazonbooks.
[6]https://tianchi.aliyun.com/dataset/53.

and the top 10% are selected as elite samples. Early stopping occurs after five iterations without improvement. Bayesian Optimization (BayesOpt) refines the CEM results by performing 10 calls (T=10 in Algorithm 1) to optimize global weights. For personalized weight assignment, optimal hyperparameters are found via grid search, with learning rates $\eta_2$ in {1e-5, 5e-5, 1e-4} and regularization weights $\lambda$ in {0.5, 1, 5}. The number of sampled weight vectors for each user $S$ in Equation (22) is set to 1. In Equation (18), $\delta_{\max} = 10.0$ and $\epsilon = 10^{-6}$ are used. Pre-trained user and item representations from SimpleX are used for initialization. The top 10 items retrieved by each channel are pooled to represent the channel, denoted as $c_{uk}$. The best models are selected based on R@200 on the validation set, and final metrics are reported on the test set.

## C.4 Hyperparameters of Baselines

We now present the hyperparameters used for the baselines across the three datasets. For ItemKNN and UserKNN, we set $k = 10$ due to the large number of both users and items. The remaining models are configured as follows: BPR: {learning rate: 5e-4}, NeuMF: {learning rate: 1e-4, MLP hidden sizes: [64, 32, 16]}, SimpleX: {learning rate: 1e-4, margin: 0.3, negative weight: 150}, and LightGCN: {learning rate: 1e-3, regularization weight: 1e-2, n layers: 3}.

## C.5 Evaluation Metrics

The metrics used in the experiments, denoted as P@L, R@L, and F1@L, are presented in the following equations:

$$\text{Precision@L} = \frac{1}{N} \sum_{u \in \mathcal{U}} \frac{|\mathcal{R}_u \cap \mathcal{T}_u|}{|\mathcal{R}_u|} \quad (27)$$

$$\text{Recall@L} = \frac{1}{N} \sum_{u \in \mathcal{U}} \frac{|\mathcal{R}_u \cap \mathcal{T}_u|}{|\mathcal{T}_u|} \quad (28)$$

$$\text{F-Measure@L} = \frac{1}{N} \sum_{u \in \mathcal{U}} 2 \times \frac{\text{Precision@L}_u \times \text{Recall@L}_u}{\text{Precision@L}_u + \text{Recall@L}_u} \quad (29)$$

## D EXTENDED ANALYSIS AND RESULTS

### D.1 Retrieval Performance and Diversity

Table 5: Evaluation of the trade-off between retrieval accuracy and diversity on Amazon_Books and Tmall.

| Model | Amazon_Books | | Tmall | |
|---|---|---|---|---|
| | R@200 | Diversity | R@200 | Diversity |
| NeuMF | 4.74% | 3.40% | 6.63% | 7.62% |
| SimpleX | 7.94% | **91.53%** | 8.72% | 76.10% |
| LightGCN | 8.29% | 53.24% | 6.84% | 19.21% |
| Equal-Weight Merging | 7.75% | 77.24% | 9.17% | 80.74% |
| BayesOpt (non-personalized) | 8.85% | 83.13% | 9.68% | **80.97%** |
| PG (personalized) | **9.21%** | 82.12% | **10.02%** | 80.81% |

There is often a trade-off between retrieval performance and diversity, yet diversity in recommendation results is crucial for user experience [37]. Various approaches [8] have been proposed to measure the diversity of the recommended list of items. We use item coverage [16, 55], which calculates the proportion of items recommended across all users, as defined in Equation (30):

$$\text{ItemCoverage} = \frac{|\bigcup_{u \in U} R_u|}{|I|} \quad (30)$$

Table 5 presents the retrieval performance and diversity of several methods. Our scientifically optimized multi-channel retrieval merging strategies achieve better retrieval performance while maintaining high diversity, effectively striking a balance.

## E DEPLOYMENT DISCUSSION

In this section, we share our hands-on experience implementing our multi-channel fusion strategy in Company X's recommendation scenario. As discussed in Section 6, the number of items retrieved from each channel varies. Given this variability, the current production system sets an upper limit on the number of items retrieved from each channel. This limitation prevents the direct application of our globally optimized weights from experimental results. Instead, we calculate adjustable ratios based on the results from CEM, ensuring they meet business requirements by truncating channels that exceed their limits and adjusting toward the optimized weights. As a result, the deployed version is an approximation of CEM, adapted to fit practical constraints.

