# OpenReview forum: "Unleashing the Potential of Multi-Channel Fusion in Retrieval for Personalized Recommendations"
_ACM.org/TheWebConf/2025/Conference — WWW 2025 Oral_

### Official Review · Reviewer_GmXQ · 2024-11-26

**Novelty:** 6
**Technical Quality:** 6

**Review:**

Summary

This paper addresses the challenge of multi-channel fusion in recommender systems' retrieval stage. The authors propose both globally unified and personalized approaches to optimally combine results from different retrieval channels. They introduce a two-stage optimization strategy using the Cross Entropy Method (CEM) and Bayesian Optimization for global weight optimization, and a policy gradient approach for personalized merging. The methods are evaluated on three large-scale datasets and deployed in a real-world system, showing significant improvements over baseline approaches.

Technical Quality

The paper demonstrates solid technical competence in algorithm design and empirical validation but could be strengthened with more rigorous theoretical analysis and scalability studies. The practical success in real-world deployment provides strong evidence for the technical soundness of the approach, though more detailed technical specifications of the deployment would be valuable. Despite these limitations, the technical quality is sufficiently high to support the paper's claims and contributions.

Clarity

The paper is well-organized and written. Key terms and background information are explained clearly in the Introduction, making it easier to follow. The diagrams and tables are well-labelled and explained, and the modular breakdown of the model helps make complex ideas more understandable. Some sections, particularly the methodology and preliminaries, are overly dense and may deter readers unfamiliar with the techniques. The technical depth may overwhelm readers unfamiliar with optimization techniques.

Originality

Similar weight optimization problems exist in other domains. The basic concept of multi-channel retrieval is not new, and some components use existing techniques (CEM, BayesOpt, PG).
However,  the approach proposed in the paper is novel and original. It claims several contributions:
 - presents the first systematic investigation of multi-channel fusion in the retrieval stage,
- a covel combination of optimization techniques (CEM + BayesOpt) for global weights,
- an innovative application of policy gradient for personalized weight assignment,
- new problem formulation for multi-channel fusion.

Significance

This work makes a meaningful contribution by solving a critical bottleneck in recommendation pipelines and advancing state-of-the-art retrieval strategies. Its real-world impact, combined with theoretical advancements, positions it as a valuable addition to the field of recommendation systems.

Strengths:

- Writing and Organization:
 -- The Introduction emphasizes the challenges of multi-channel retrieval and its importance in recommender systems.
 --It identifies three key issues in multi-channel fusion: reliance on heuristic methods, sensitivity to weight combinations, and the non-differentiable nature of the selection process.
 -- The authors effectively justify their study's relevance by highlighting these gaps.
 --The background section thoroughly explains the multi-stage cascade ranking system, distinguishing between retrieval and ranking stages.

- Approach and its Evaluation
--The use of Cross Entropy Method (CEM) and Bayesian Optimization for global weight optimization is innovative, providing a robust solution to the non-differentiable nature of the problem.
--The personalized weight assignment, based on policy gradient approaches, demonstrates the model's adaptability to user-specific preferences.
--The detailed pseudocode and visualizations effectively clarify the proposed strategies in the methodology section.
--The extensive experiments validate the proposed methods using three large-scale datasets (Gowalla, Amazon_Books, and Tmall).
--The results demonstrate significant improvements over baseline models in precision, recall, and F-measure.
--The inclusion of real-world deployment data (and offline and online evaluation) further strengthens the paper's practical relevance.

Weaknesses:

- Some sections, particularly the methodology and preliminaries, are overly dense and may deter readers unfamiliar with the techniques.
- Lacks quantitative evidence for why current industry practices are suboptimal. In other words, the impact of suboptimal fusion is stated but not quantified.
- There is limited discussion of why existing solutions are insufficient. For example, the paper mentions that “These methods lack systematic analysis, leading to suboptimal performance” or “existing simple quota mechanisms are inflexible and fail to accommodate personalization”, but does not discuss the main reason. How is the term “inflexible” defined?
- Limited discussion of related work in fusion techniques.
- The technical depth may overwhelm readers unfamiliar with optimization techniques.
- Lacks a detailed discussion of limitations and potential biases in the experiments.
- The paper uses policy gradient (PG) for personalized weight assignment but doesn't discuss:
   -- Whether the training process consistently converges
   -- How many iterations are typically needed
   -- If there are any guarantees of finding optimal/near-optimal solutions
   -- What conditions are necessary for convergence
   -- Whether different initializations lead to similar results.

**Questions:**

1. How does the computational complexity of your approach scale with the number of retrieval channels? What is the maximum number of channels that can be practically supported?

2. The paper claims to address the cold-start problem; you pad the shorter retrieval channels to match the longest one, aligning with your problem formulation. How is this padding done?

3. In the real-world deployment section, you mention engineering limitations that prevented the implementation of the personalized approach. What are these limitations, and could you elaborate on these specific challenges and potential solutions?

4. Could you provide more details about the convergence properties of your policy gradient approach? How do you ensure stable training?
What specific limitations might affect the scalability of this method in environments with significantly larger datasets or more channels?

**Ethics Review Description:**

A preprint is available on arXiv (2410.16080), which is not anonymized; the submission does not meet the double-blind review requirement.

**Ethics Review Flag:**

Yes

**Reviewer Confidence:**

3: The reviewer is confident but not certain that the evaluation is correct

**Scope:**

4: The work is relevant to the Web and to the track, and is of broad interest to the community

---

### Official Review · Reviewer_dR2J · 2024-11-28

**Novelty:** 5
**Technical Quality:** 6

**Review:**

Summary

The paper addresses the problem of multi-channel fusion in recommender systems, introducing novel strategies for assigning optimal weights to each channel. To do this, they use back-box optimization techniques, such as Cross Entropy Method and Bayesian Optimization for global weight optimization (non-personalized weight assignment) and policy gradient approaches for personalized merging. Experiments conducted on three datasets show the superiority of the proposed method over state-of-the-art approaches. Moreover, the authors also deploy the proposed approach in a real-world scenario.

Evaluation:

Strengths:

- The paper is well-written, easy to follow, and technically sound.
- Deployment in real-world scenarios; results demonstrate the successful deployment of the proposed approach (just the non-personalized version) and its superiority over the strategy employed by the company.

Weakness:
- There is a lack of discussion about the computational efficiency of the proposed approaches.

- Due to current engineering limitations, personalized multi-channel fusion was not implemented in the real-world deployment. This raises questions about the practicality of adopting such an approach in industry settings.

- Although the proposed approaches were tested on three datasets and deployed in a real-world scenario to show their generalizability, the authors should consider testing them by using other well-known and used datasets in recommender systems (especially to show the generalizability of the personalized version that was not deployed in a real-world setting).

**Questions:**

Table 2 reports the precision, recall, and f1 value at L=200. Why did you choose 200? I suggest analyzing different values of L. Also, in section 5.1, you stated that you randomly selected 100 negative samples for Gowalla and Amazon Books and 200 for Twall. Also, in this case, why did you choose these numbers exactly? Have you performed some preliminary analysis that guided your decision?

**Reviewer Confidence:**

3: The reviewer is confident but not certain that the evaluation is correct

**Scope:**

3: The work is somewhat relevant to the Web and to the track, and is of narrow interest to a sub-community

---

### Official Review · Reviewer_QawU · 2024-11-30

**Novelty:** 4
**Technical Quality:** 4

**Review:**

This paper presents a novel approach to multi-channel fusion in recommender systems' retrieval stage, addressing a significant gap in existing research. The authors propose two key methodologies: a two-stage optimization strategy combining Cross Entropy Method and Bayesian Optimization for global weight assignment, and a policy gradient-based approach for personalized channel fusion. The work effectively addresses the non-differentiable nature of channel selection through black-box optimization techniques. Extensive experiments on three large-scale datasets and real-world deployment demonstrate substantial improvements over baseline methods, with up to 10.43% improvement in Recall@200.

Strengths:

(1) First, this paper introduces a systematic approach to multi-channel fusion, moving beyond traditional heuristic methods and establishing a formal framework for optimization in the retrieval stage of recommender systems.

(2) This paper provides empirical validation through experiments across multiple large-scale datasets and successful real-world deployment, providing robust evidence for the effectiveness of their proposed methods.

(3) The experimental results demonstrates the improvements in recommendation performance.

Weaknesses:

(1) First, while the authors claim to be the first to systematically investigate multi-channel fusion, their methodological innovation is somewhat incremental. The proposed approaches largely combine existing techniques (Cross Entropy Method, Bayesian Optimization, and policy gradient) rather than introducing fundamentally new algorithms.

(2) Second, the comparison baseline of this article is relatively old. It is recommended to add the most recent baseline method to enhance the persuasiveness of the article..

(3) Third, the experimental validation, although conducted on three datasets, lacks diversity in application scenarios. All datasets are focused on user-item interactions in e-commerce. The absence of experiments in other domains such as news recommendation, video streaming, or social networks limits the generalizability of their findings.

(4) Fourth, the authors fail to thoroughly analyze the computational complexity and efficiency of their proposed methods, particularly for the personalized approach.

**Questions:**

Please see the weaknesses.

**Reviewer Confidence:**

3: The reviewer is confident but not certain that the evaluation is correct

**Scope:**

4: The work is relevant to the Web and to the track, and is of broad interest to the community

---

### Official Review · Reviewer_2PLS · 2024-12-01

**Novelty:** 4
**Technical Quality:** 5

**Review:**

This paper addresses the critical issue of multi-channel fusion in the retrieval stage of recommender systems. The authors identify key challenges in merging diverse results from multiple candidate generators, such as sensitivity to weight combinations and the non-differentiable nature of the selection process. To tackle these challenges, the paper proposes a two-part approach: (1) globally unified weight assignments using black-box optimization techniques, and (2) personalized weight assignments leveraging policy gradient methods from reinforcement learning. Extensive experiments on real-world datasets demonstrate the proposed methods' superiority over existing baselines.

Pros:
- The paper is well-motivated, focusing on the practical and critical problem of improving retrieval quality by effectively fusing multi-channel information. This is a significant challenge in real-world recommender systems.
- The two-stage black-box optimization strategy and policy gradient-based personalized merging approach are well-motivated and innovative.
- Experiments demonstrate the effectiveness of the proposed method. Both offline and online results varify the ultility.

Cons:
- The selected candidate generator methods mainly focus on recommendation methods, lack the diversity of retrival results.
- Current SOTA SSL-based methods are not included, such as SimGCL. Besides, the authors should give a more clearer illustration of merging process.

**Questions:**

Please see weaknesses.

**Reviewer Confidence:**

2: The reviewer is willing to defend the evaluation, but it is likely that the reviewer did not understand parts of the paper

**Scope:**

3: The work is somewhat relevant to the Web and to the track, and is of narrow interest to a sub-community

---

### Official Review · Reviewer_LPdM · 2024-12-02

**Novelty:** 6
**Technical Quality:** 6

**Review:**

The paper studies multistage recommender systems, and specifically the use of multiple ranking methodologies to retrieve and rank items. Weights are used to determine the impact level of each ranking method, and multiple novel approaches based on optimisation are proposed to find the weights. The writing, experiments, and description of the problem setting are clear and precise, and the paper gives appropriate motivation for the optimization problem it proposes.

Pros:
1) Writing, tables, appendices and technical details are of high quality.
2) The work proposes a new direction which seems to be a lacuna in the literature.
3) The empirical results are strong and consistent, and sufficient experiments were carried out with multiple baselines.

Cons:
1) The multistage retrieval setup is motivated by latency concerns, and it would be nice to see some kind of quantitative comparison of the computational cost of different methods.

**Questions:**

1) As noted in the paper, the optimized weight distributions assign very low weight to certain channels. After optimization, if removing those channels does not hinder performance it would seem to be beneficial to remove them entirely to reduce the complexity and latency of the overall system. Were any experiments of this type carried out?

**Reviewer Confidence:**

3: The reviewer is confident but not certain that the evaluation is correct

**Scope:**

4: The work is relevant to the Web and to the track, and is of broad interest to the community